# Comparative Transcriptome Analysis Reveals the Effects of a High-Protein Diet on Silkworm Midgut

**DOI:** 10.3390/insects16040337

**Published:** 2025-03-24

**Authors:** Xinyi Chen, Jiahao Li, Yuxi Shan, Qiaoling Wang, Pingzhen Xu, Heying Qian, Yangchun Wu

**Affiliations:** 1Jiangsu Key Laboratory of Sericultural and Animal Biotechnology, School of Biotechnology, Jiangsu University of Science and Technology, Zhenjiang 212100, China; 221211801101@stu.just.edu.cn (X.C.); 221111802128@stu.just.edu.cn (J.L.); 231211802109@stu.just.edu.cn (Y.S.); 241211803107@stu.just.edu.cn (Q.W.); xpz198249@just.edu.cn (P.X.); qhysri@just.edu.cn (H.Q.); 2Key Laboratory of Silkworm and Mulberry Genetic Improvement, Ministry of Agriculture and Rural Affairs, Sericultural Scientific Research Center, Chinese Academy of Agricultural Sciences, Zhenjiang 212100, China

**Keywords:** silkworm, high-protein diet, midgut, metabolism, transcriptome

## Abstract

Protein plays a significant functional role in the silkworm diet. However, the impact of the high-protein diet (HPD) on silkworm growth and development has not been extensively studied. Our research found that an HPD 6% concentration promoted silkworm development. We screened genes related to the majority pathway of mitochondrial oxidative phosphorylation, ribosome, and ribosome biogenesis in eukaryotes and asnalyzed their expression in the midgut tissue. HPD 6% treated increased ATP levels and enhanced the activities of SOD and NADH.

## 1. Introduction

The domesticated silkworm (*Bombyx mori*) is a model insect of Lepidoptera and is widely used in basic and applied research [1]. The silkworm is an oligophagous insect. Mulberry leaves are its sole diet [2]. About 20 g of mulberry leaves are ingested and digested by silkworm larvae, and the larva grows to 5 g in the wandering stage. The nutrients are obtained from the leaf used by the silkworm larvae for body growth, development, and cocoon formation [3]. In the fifth instar of silkworm larvae, 85% of the total leaves are consumed.

The components of mulberry leaves used by silkworms mainly include protein, carbohydrates, lipids, inorganic substances, moisture, and vitamins [4]. The silkworm digestive system is a simple tube divided into the fore-, mid-, and hindgut according to its function and structure [5], of which the midgut comprises approximately 78 percent of the total length of the digestive tube and is the most important section [3]. The midgut comprises muscle, a basement membrane, an epithelium layer, and a peritrophic envelope [6]. The midgut of the silkworm plays a crucial role in digestion, absorption, and nutrient metabolism [4,7].

Most insects probably need optimum protein content in their diet for the best growth. Dietary protein is crucial for insect survival, development, and reproduction [8,9]. Protein can control and adjust the substance for the metabolism and physiological function of the silkworms by combining active substances [4]. A high-protein diet enhances insect digestion, absorption, and metabolism [10,11]. A low-protein diet inhibits insect protein synthesis for energy metabolism [12]. Elevated dietary protein intake increases basal levels of immune activity in insects [13,14].The quantity of dietary protein ingested after emergence can positively influence worker longevity, that is, due to the increased antioxidant gene expression in honey bees [15]. Dietary protein is a precursor to amino acids (AAs) that proficiently modulate glucose and lipid metabolism throughout the body [16,17]. High-protein diets enhance energy expenditure, reduce blood glucose levels, and facilitate fat oxidation [18,19]. Milk is a complex liquid that provides large amounts of protein to promote healthy growth and development. Silkworm larvae can be fed mulberry leaves treated with bovine milk for better growth and increased silk production [20]. Skim milk powder has a high protein quality and a low fat content, which helps reduce lipid accumulation [21,22]. Considering the beneficial properties of skim milk powder, we added skim milk powder to the diet of silkworms.

This work first focused on a comprehensive analysis of silkworm gene expression under increased protein intake conditions. The results indicated modifications in oxidative phosphorylation and ribosomal pathways induced by changes in protein consumption. For silkworms, the activation of metabolic pathways in the midgut likely represents one of the key factors contributing to improved digestive capacity and development during later growth stages due to excessive protein intake, underscoring the role of the midgut as a metabolic regulator.

## 2. Materials and Methods

### 2.1. Insect

The silkworm 871 strain was provided by the Sericulture Research Institute of the Chinese Academy of Agricultural Sciences, Zhenjiang, China, and reared during the spring. Eggs were incubated at 25–26 °C until larvae hatch. The larvae were reared under standard conditions at 25–26 °C and 65–70% humidity. The larvae were fed fresh mulberry leaves until the 1st day of the 5th instar (I5D1).

### 2.2. Protein Treatment and RNA Preparation

Skimmed milk powder (Yili, China) was used. The nutrition facts per 100 g were as follows: 35.2 g protein, 55.2 g carbohydrates, 1592 KJ energy, 420 mg sodium, 1200 mg calcium, 0.010 g cholesterol, and 1.5 g fat. In the pre-experiment regarding the survival rate effects, 0 (control), 30 (HPD 3%), 60 (HPD 6%), 120 (HPD 12%), 180 (HPD 18%), and 240 (HPD 24%) g of L-1 skimmed milk powder were used to expose the 5th instar larvae (I5D1). All mulberry leaves were selected to ensure a similar state. The petioles and stems of the leaves were removed, and the leaves were cut into equal sizes. The leaves were fully submerged in the solution for 15 minutes, guaranteeing uniform coverage of each leaf. The leaves were subsequently removed for natural drying until no visible liquid remained. The silkworm larval density was regulated to ensure each larva had adequate access to the leaves. The control group and treatment group with the high-protein diet (HPD) were established by each group, with three replicates consisting of 150 silkworms per replicate. The control group was fed mulberry leaves soaked in deionized water, while the treatment group with HPD was fed mulberry leaves soaked in skimmed milk powder solutions.

The midgut was collected from the control group and the HPD 6% group. The total RNA of the sample was extracted using the Beijing Aide-Lai Biotechnology Co., Ltd.’s (Beijing, China) EASYspin Plus rapid tissue/cell RNA extraction kit. The RNA concentration, purity, and integrity were assessed using the Nanodrop 2000 Bioanalyzer (Thermo Fisher Scientific, Waltham, MA, USA).

### 2.3. Library Construction and Illumina Hiseq Xten Sequencing

Total RNA was extracted from I5D3 midgut tissues, and ribosomal RNA was depleted to isolate the mRNA. The mRNA is then fragmented using divalent cations in the NEB Fragmentation Buffer following the standard NEB protocol [23]. Utilizing the fragmented mRNA as a template and random oligonucleotide ides as primers, an M-MuLV reverse transcriptase system synthesized the first cDNA strand. Subsequently, the RNA strand was degraded by RNaseH, and the second cDNA strand was synthesized in a DNA polymerase system using dNTPs as substrates. The purified double-stranded cDNA underwent end repair, A tailing, and ligation of the sequencing adapters. After the library construction, initial quantification was performed with a Qubit2.0 Fluorometer, and the library was diluted to 1.5 ng/µL. Then, the insert size of the library was tested using an Agilent 2100 bioanalyzer (Agilent Technologies, Santa Clara, CA, USA). After a qualified library check, different libraries were pooled according to the requirements of effective concentration and target data volume for Illumina sequencing.

### 2.4. Read Mapping

Under the Linux operating system, rRNA sequences are extracted using SEQKIT (v2.5.1) software, and rRNA indexes are constructed and compared using HISAT2 software (http://daehwankimlab.github.io/hisat-genotype/, accessed on 6 September 2023) to remove mismatched sequences and reduce the proportion of rRNA in the output [24]. This process reduced bias towards RNA sequencing and assembly. Three essential parameters are assessed using the FastQC online tool (https://www.bioinformatics.babraham.ac.uk/projects/fastqc/, accessed on 10 September 2023) [25], particularly Q20 (representing the percentage of bases with a mass fraction of 20 or more), Q30, and GC content, to assess the quality of all reads. HISAT2 (v2.1.0) software was utilized for fast and accurate detection.

### 2.5. Differential Expression Analysis

The analysis of differential expression genes (DEGs) required standardizing gene expression data from different samples to eliminate differences in sequencing depth and gene length between the samples. Thus, gene expression was normalized through fragments per kilobase of exon model per million mapped fragments (FPKM). The resulting *p*-values were accustomed to following Benjamini and Hochberg’s method for regulating the false discovery rate (FDR). Genes were read by RSEM (v1.3.3) software. The DEGs were screened by DESeq2 (v1.46.0) software, with the conditions set at |log_2_(FoldChange)| > 0 and *p*-value < 0.05.

### 2.6. Functional Annotation and Enrichment of DEGs

To delve deeper into the biological roles of DEGs, we performed a comprehensive annotation process using two key bioinformatics databases, Gene Ontology (GO) and the Kyoto Encyclopedia of Genes and Genomes (KEGG). GO functional enrichment analysis is performed to identify GO terms and metabolic using Goatools (v0.6.5) [26]. A statistical threshold *p*-value < 0.05 was applied to determine the significance of these enrichments.

### 2.7. Gene Set Enrichment Analysis (GSEA)

To identify genes that might not be significantly different in overall expression levels but were biologically significant, we performed a gene set enrichment analysis (GSEA). Pathways with *p*-value < 0.05, false discovery rate (FDR) < 0.25, and normalized enrichment score |NES| > 1 were selected for further analysis.

### 2.8. Expression Validation of DEGs from RNA-Seq by qRT-PCR

Energy metabolism-related genes were selected for validation by quantitative real-time polymerase chain reaction (qRT-PCR), and the extracted total RNA (1 μg) was reverse transcribed into cDNA using a 5× Hifiscript RTMaster (Kangwei, Nanjing, China). *BmGAPDH* was selected as the internal reference gene, and the data were analyzed using the instrument’s software, Light Cycler^®^ 96 SW 1.1. The primers are shown in Appendix A.

### 2.9. Assessment of Total Nitrogen (TN) in Silkworm Feces and Enzyme Activity Assay

Silkworm feces were collected from both groups two and three hours after feeding on I5D3 and I5D5. The midguts, malpighian tubules, middle silk glands, and posterior silk glands were collected from the HPD and control groups on I5D3 and I5D5, respectively. Dumas combustion was used to determine the total nitrogen (TN) content of the silkworm feces. The activities of superoxide dismutase (SOD), catalase (CAT), and nicotinamide adenine dinucleotide (NADH and NAD+) in the midgut were detected using commercial assay kits (Solarbio, Beijing, China). An ATP test kit (Solarbio, Beijing, China) detected ATP levels in the midgut, malpighian tubule, middle-silk-gland, and posterior-silk-gland.

### 2.10. Data Processing

The data were analyzed with a one-way ANOVA by Prism 8.0 (GraphPad Prism Inc., San Diego, CA, USA). Relative mRNA expression was calculated using the 2^−ΔΔCt^ comparative CT method, and Students’ *t*-test was applied to compare the statistical difference with Prism 8.0.

## 3. Results

### 3.1. Determination of HPD

The increase in protein concentration is associated with a rise in the mortality rate of silkworms. Meanwhile, the investigation period of the mortality rate extended from the first day of rearing to the adult stage. The mortality of silkworms exhibited a concentration-dependent response to HPD (3–24%) when compared to the control group (Appendix A). The mortality rate increased to 19.67% at an HPD 3% concentration. With a slight increase to HPD 6%, the mortality rate rose to approximately 21.66%. Further escalation of the HPD concentration resulted in progressively higher mortality rates: 25% at HPD 12%, 37% at HPD 18%, and 47% at HPD 24%, suggesting that excessive protein intake might be detrimental to their development. The mortality rates of HPD 3% and HPD 6% were not significant compared to the control group. After HPD 12%, the mortality rate was significantly higher than that of the control group. Based on these findings, HPD 6% and HPD 12% were used for the next investigation.

### 3.2. Effects of HPD on Larval Weight and Economic Character of Cocoon

The larval weight of each group (twenty female silkworms) was recorded daily (Appendix A). The larval weight of silkworms in the HPD 6% group was generally higher than that of the control group. The weight of the HPD 12% dietary treatment group was lower than the control group, except for a transient increase observed on the second day. At other times, the weight was not significantly higher than the control group, and it was significantly lower after five days of dietary supplementation. The time mature silkworms appeared in the HPD groups was 24 h earlier than in the control groups (Figure 1A). The total nitrogen content in I5D3 silkworm feces collected from the HPD 6% and HPD 12% was markedly higher than that of the control group. In contrast, the content was basically the same between the HPD 6% group and the control group on I5D5. However, HPD 12% was significantly higher than that of the control group (Figure 1B). The body size of the HPD 6% group was significantly larger than that of the control group, whilst the body size of the HPD 12% group was comparable to that of the control group (Figure 1C). HPD 6% was conducive to the digestion of nitrogen within silkworms, strengthened the digestive capability of the midgut, and benefited the increase in silkworm body mass. Nevertheless, in the HPD 12% group, a nitrogen imbalance might exist within silkworms, which was detrimental to their growth. The weights of the cocoon shell and pupa were recorded six days after mounting the silkworms for spinning. The cocoon shell rate was determined using the formula: cocoon shell weight (g)/cocoon weight (g) × 100%. Fifteen female cocoons were chosen from each zone for examination and replicated three times. The cocoon weight, pupal weight, and cocoon shell rate of the HPD 6% group were considerably higher than that of the control group, whilst HPD 12% was not significantly different from the control group (Figure 1D–G). Consequently, a concentration of HPD 6% was selected for subsequent transcriptome analysis.

### 3.3. Transcriptome Profiling of DEGs

The valid data for Q20 and Q30 exceeded 95%, indicating high data quality (Appendix A). Pearson’s correlation coefficient was employed to assess the similarity among the various samples. An inter-sample Pearson correlation analysis demonstrated that the correlation coefficients within each of the six sample groups exceeded 90%, signifying a substantial degree of gene expression similarity among the samples and variability between the groups (Figure 2A). Upon comparison of the readings with the genome, 12,526 genes were discovered, including 116 new genes. A control and protein-treated midgut comparison revealed 1724 significantly changed genes, comprising 803 up-regulated genes and 921 down-regulated genes (Figure 2B).

### 3.4. Weighted Correlation Network Analysis (WGCNA)

WGCNA analysis was conducted to discern differential gene expression patterns. No anomalous samples were identified according to the average clustering analysis (Figure 3A). The soft threshold β was established at 16 to detect scale-free networks (Figure 3B). Genes were categorized into 25 modules, with the grey module comprising genes that could not be clustered (Figure 3C). Three gene modules were discovered that had a strong correlation between the HPD 6% group (Figure 3C, red, brown, and pink) and the control group (Figure 3C, black, indigo, and green). Hub genes within these modules were analyzed using Cytoscape (v3.9.1) software. The CytoHubba application was used to analyze the extracted data obtained by three calculation methods. The top 10 hub genes screened by maximal clique centrality (MCC), maximum neighborhood component (MNC), and edge percolated component (EPC) algorithms were all ribosomal genes, and eight genes were identical among them (Figure 3D).

### 3.5. Functional Annotation and Enrichment Analysis of DEGs

The GO annotation of DEGs was independently analyzed and categorized into three classes: biological process (BP), cellular component (CC), and molecular function (MF). The top 30 subclasses of GO were illustrated (Figure 4A). In the BP category, the most prevalent sub-classes were biosynthetic and metabolic processes. In the CC category, most DEGs were associated with the ribosome, ribonucleoprotein complex, mitochondrion, and cytoplasm. Within the MF category, which contained the most significant number of DEGs, the main three sub-classes were structural constituent of ribosome, ATP binding, and adenyl nucleotide binding. Among the GO functional enrichment analysis terms, there were associations with ATP binding and ATPase activity (Figure 4B).

The up-regulated and down-regulated genes of DEGs were separately analyzed in the KEGG pathway annotation (Figure 4C). The up-regulated genes exhibited the majority pathway of oxidative phosphorylation, ribosome, and ribosome biogenesis in eukaryotes. The down-regulated genes of DEGs were mostly annotated in ABC transporters, lysosome, endocytosis, and sphingolipid metabolism pathways.

We performed gene set enrichment (GSEA) analysis using the KEGG-based list to enrich the gene sets. The significant upregulation of ten gene sets in the HPD 6% group was related to mitochondrial energy production and nuclear transcription (Appendix A). One pathway was oxidative phosphorylation (bmor00190), nine pathways involved in nuclear transcription were DNA replication (bmor03030), ribosome (bmor03010), ribosome biogenesis in eukaryotes (bmor03008), aminoacyl-tRNA biosynthesis (bmor00970), spliceosome (bmor03040), base excision repair (bmor03410), mismatch repair (bmor03430), nucleotide excision repair (bmor03420), and fanconi anaemia pathway (bmor03460). The GSEA plots enriched for key pathways are shown in Figure 4D. The ten gene sets (pathways) detected by GSEA-based KEGG overlapped with those based on DEG.

### 3.6. HPD 6% Activation Oxidative Phosphorylation and Nuclear Transcription

The mitochondrial electron transport chain (ETC) consists of five protein complexes integrated into the inner mitochondrial membrane that utilize a series of electron transfer reactions to generate cellular ATP through oxidative phosphorylation (Figure 5A). The mitochondrial oxidative phosphorylation (OXPHOS) exhibited considerable enrichment in the HPD 6% midgut. In the midgut of the HPD 6% group, the expression levels of fifty-eight genes were up-regulated across each of the five complexes (Figure 5B, Appendix A), with eighteen genes in complex I, two genes in complex II, ten genes in complex III, thirteen genes in complex IV, and fifteen genes in complex V and only one gene (*Ndufa6*) in complex I down-regulated. The expression levels of the six genes (*ND5*-*2*, *Nd13*-*B*, *COX5B*, *COX2*, *ATPsynCF6*, and *Vha100*-*2*) selected from the OXPHOS pathway were consistent with the results of the RNA-seq data (Figure 5C). Accelerated energy consumption led to inevitable oxidative stress in the organisms. Hence, we examined the activities of antioxidant-related enzymes. The activities of SOD, NADH, and NAD+ in the midgut of the HPD 6% group were significantly higher than that in the control group on I5D3 and I5D5. In contrast, the activity of CAT was decreased considerably (Figure 5D). The ATP levels in the midgut, malpighian tubule, middle silk gland, and posterior silk gland of the HPD 6% group were significantly higher than that in the control group on I5D3 and I5D5 (Figure 5D).

Ribosomes are the cellular factories responsible for making proteins (Figure 5E). In eukaryotes, ribosome biogenesis involves the production and correct assembly of four rRNAs and about 80 ribosomal proteins. We observed that eighty-eight enriched genes had up-regulated expression in the pathways of the ribosome (nineteen genes in small subunit and thirty-one genes in large subunit), ribosome biogenesis in eukaryotes (twenty-five genes), base excision repair, mismatch repair, and DNA replication (total of thirteen genes in the three pathways of base excision repair, mismatch repair, and DNA replication). Only one gene (*PARG*) in base excision repair was down-regulated (Figure 5F, Appendix A).

## 4. Discussion

The silkworm is a well-studied Lepidopteran model system for its life cycle and economic importance. It is also susceptible to nutritional factors. Dietary levels affect the growth rate, developmental period, body weight, larvae survival rate, silk production, longevity, and the fecundity of adults. Silkworm larvae eat voraciously at the fifth-instar period and represent maximum growth among all its larval stages. Day 3 of the fifth-instar larvae is the boundary for the larval stage, and the larvae then greatly synthesize silk proteins in the silk gland [3].

In this study, HPD 6% feeding had a positive effect on the growth of fifth-instar larvae, resulting in a better growth rate and increased silk production. The results of an RNA-seq showed that 7202 genes were discovered, and 1724 genes were significantly changed, comprising 803 up-regulated genes and 921 down-regulated genes. The up-regulated genes exhibited the majority pathway of mitochondrial oxidative phosphorylation, ribosome, and ribosome biogenesis in eukaryotes. The down-regulated genes were mostly annotated in ABC transporters, lysosome, endocytosis, and sphingolipid metabolism pathways. Endocytosis transforms food into food vacuoles via surface receptors and membrane proteins, and lysosomes fuse with food vacuoles for the digestion of food [27]. Our results indicated that in pathways associated with endocytosis and lysosome, most down-regulated genes were primarily related to apoptosis and lysosomal acid hydrolases. In contrast, up-regulated genes were enriched in membrane transport proteins and vesicular or vacuolar ATPases. V-ATPase can rapidly modulate the pH of endosomes and/or lysosomes, thereby maintaining the normal functionality of lysosomes [28]. This indicated that increased protein levels might disrupt the lysosome’s acidic environment, leading to the regulation of lysosomal acid hydrolases to restore the acidic environment necessary for normal lysosomal function. As the dietary protein content increased, *eIF4E* in the mTOR pathway was up-regulated, leading to an acceleration in the rate of protein synthesis. The activities of SOD, NADH, and NAD+ in the midgut of the HPD 6% group were significantly higher than that in the control group on I5D3 and I5D5. The total nitrogen content in I5D5 silkworm feces collected from the HPD 6% group was the same as that of the control group. These results indicated that the digestive capability of fifth-instar silkworm larvae with HPD 6% could be improved.

In mouse models [29], a similar study explores the effects of protein supplementation, yielding results similar to those observed in our research, such as increased body weight and increased ability to cope with oxidative stress. Interestingly, researchers have attributed these benefits to abundant beneficial gut microbiota and a reduction in harmful bacteria [29]. In comparison, our study primarily focused on transcriptome pathways, providing a different perspective than oxidative phosphorylation and nuclear transcription. However, the role of the microbiota presents a compelling avenue for further investigation, offering valuable insights that can complement our approach. A high-protein diet decreases oxidative stress and increases mitochondrial respiration and mitochondrial structural components [30,31]. The mitochondria convert nutrients, including sugars, fats, and amino acids, into ATP via oxidative reactions, which are primarily regulated by the mitochondrial respiratory chain comprising complexes I–V [32]. In the oxidative phosphorylation pathway of mitochondria, Complex I functions as the primary electron acceptor during aerobic respiration chains, facilitating the oxidation of NADH. Notably, Complex I stands as the most substantial OXPHOS complex, encompassing a total of 45 subunits [32,33,34]. In our study, we observed that the expression levels of 18 NADH genes were up-regulated, and the activity of NADH was significantly enhanced in the midgut of the HPD 6% group, implying a higher energy requirement. However, on the high-fat diet, impairment of mitochondrial respiration at the level of Complex I and decreased ATP content are observed in *Drosophila melanogaster* [35]. Complex II is shared between the tricarboxylic acid (TCA) cycle and the mitochondrial electron transport chain (ETC) [32]. Complex II comprises four nDNA-encoded subunits, and it functions through the independent maturation of SDHA, SDHB, and SDHC + SDHD mediated by subunit-specific chaperones [36]. SDHB incorporates its Fe–S clusters [37]. SDHD is one of the constituent members of the hydrophobic subunits, which constitutes the Complex II membrane anchor, containing a haem b group and two CoQ binding sites [38,39]. In our result, the SDHD and SDHB expression levels were up-regulated in the midgut of the HPD 6% group, indicating the TCA cycle’s function and efficiency enhanced. Additionally, the expression levels of ten genes in Complex III, thirteen genes in Complex IV, and fifteen genes in Complex V were up-regulated in the midgut of the HPD 6% group. This increase in OXPHOS capacity could be stimulated by acetyl-CoA, which is one of the final catabolic products of amino acid metabolism. Since acetyl-CoA is directly consumed by mitochondria through the TCA cycle, HPD 6% enhanced OXPHOS. We concluded that HPD 6% decreased oxidative stress and increased mitochondrial activity in the silkworm midgut.

Ribosomes are central to protein synthesis and convert transcribed mRNAs into polypeptide chains [40]. In eukaryotes, ribosome biogenesis involves producing and correctly assembling four rRNAs and about 80 ribosomal proteins to make ribosomes, which are essential for cell proliferation, differentiation, apoptosis, development, and transformation [41]. Conflicts between the replication and transcription machinery are widespread in both prokaryotes and eukaryotes, and they both can cause DNA damage and compromise the complete, faithful replication of the genome [42]. During exposure to imidacloprid, ribosomes are upregulated, revealing that protein synthesis and metabolism increased with the demand for energy supply [43]. In this study, we observed that eighty-eight enriched genes had up-regulated expression in the ribosome (nineteen genes in a small subunit, and thirty-one genes in a large subunit), ribosome biogenesis in eukaryotes (twenty-five genes), base excision repair, mismatch repair, and DNA replication (total of thirteen genes in the three pathways), and one gene was down-regulated in base excision repair. These indicated an increasing demand for protein synthesis and metabolism, resulting in a higher mitochondrial translation rate. Protein synthesis is expedited during this process, and the ATP consumption rate is elevated [44,45,46]. Consequently, the heightened mitochondrial function facilitated the metabolism of HPD 6% components in the midgut. Meanwhile, the complex V synthases VATP supplied energy for cellular development. Fluoride exposure severely induced the damage of oxidative phosphorylation, leading to the inhibition of ATP production [47]. In our study, the ATP levels in the midgut, malpighian tubule, middle silk gland, and posterior silk gland of the HPD 6% group were significantly higher than that in the control group on I5D3 and I5D5. These results indicated that the digestion, absorption, and metabolism functions of silkworms were enhanced in vivo. The growth rate of silkworms exhibited acceleration, which aligns with the observed performance of *Manduca sexta* and *Spodoptera littoralis* when subjected to a high-protein diet [48,49]. The silkworm exhibits a robust capacity for protein uptake and utilization [50].

## 5. Conclusions

This study demonstrated that HPD 6% promoted growth and silk production in fifth-instar silkworm larvae. RNA-seq analysis revealed the upregulation of mitochondrial oxidative phosphorylation, ribosome biogenesis in eukaryotes, and ribosome pathways, whilst downregulating the ABC transporters, lysosome, endocytosis, and sphingolipid metabolism pathways. Treatment with HPD 6% enhanced mitochondrial activity and reduced oxidative stress, accompanied by increased ribosomal activity and DNA repair capacity in the midgut. These alterations accelerated protein synthesis and ATP consumption, with elevated ATP levels observed in the midgut, malpighian tubule, and silk glands. The treatment significantly increased SOD and NADH activities in the midgut compared to those in the control group. These findings provide novel insights into the molecular mechanisms underlying the nutritional regulation of silkworm development in fifth-instar larvae.

## Figures and Tables

**Figure 1 insects-16-00337-f001:**
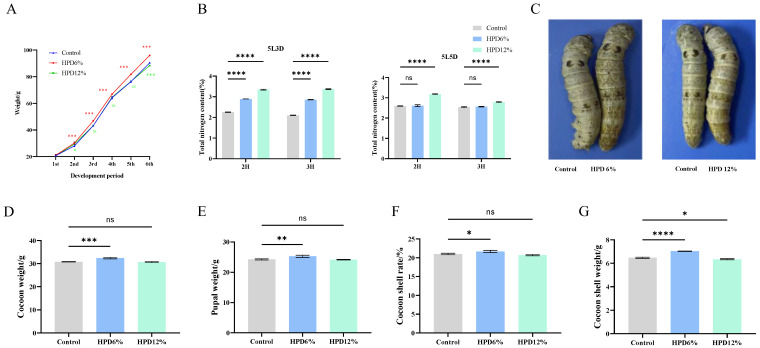
Investigation of larval weight and economic traits after HPD supplementation. (**A**) The weight of 20 female silkworms between control and HPD treatment group. The horizontal coordinate represented the development period; (**B**) total nitrogen in silkworm feces on I5D3 and I5D5. 2H: two hours, 3H: three hours; (**C**) silkworm phenotype; (**D**) cocoon weight; (**E**) pupal weight; (**F**) cocoon shell weight; (**G**) cocoon shell rate. The data were subjected to a *t*-test using GraphPad Prism (v9.0). Each group of data was tested three times, with “*” indicating a significant difference (*p* < 0.05), “**” indicating an extremely substantial difference (*p* < 0.01), “***” indicating an extremely significant difference (*p* < 0.001), “****” indicating an extremely significant difference (*p* < 0.0001), “ns” stands for not significant.

**Figure 2 insects-16-00337-f002:**
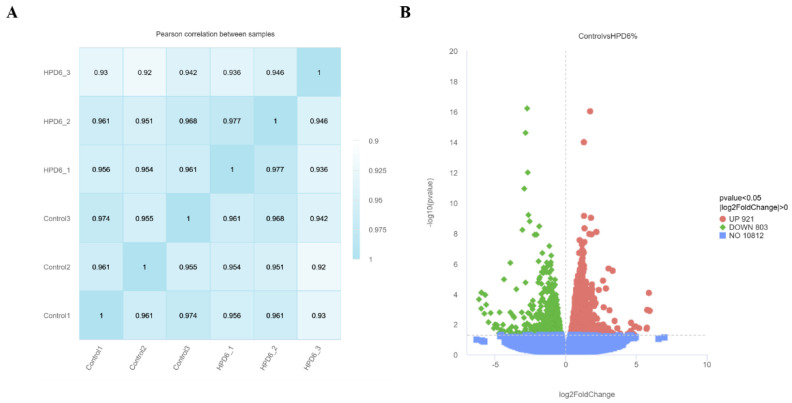
Correlation analysis and volcano gram. (**A**) Correlation analysis. HPD 6_1, HPD 6_2, and HPD 6_3 represented the three replicates of the HPD 6% group. Control1, Control2, and Control3 represented the three replicates of the control group. (**B**) The volcano plot of DEGs. The red, green, and blue dots indicated significantly up-regulated, significantly down-regulated, and non-significantly expressed genes.

**Figure 3 insects-16-00337-f003:**
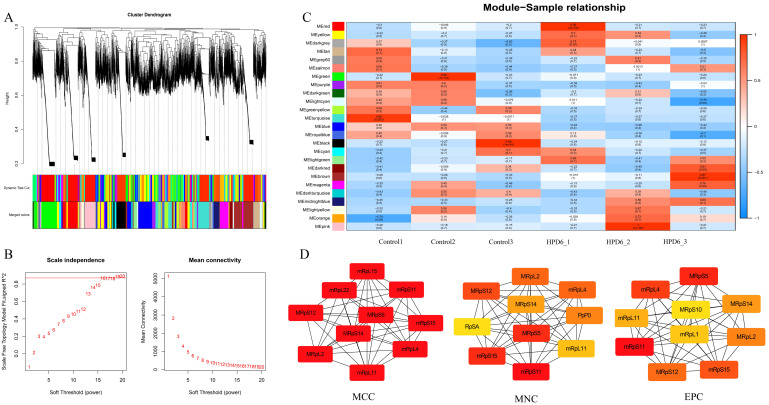
Identification of subtype-specific gene profile and biological function by WGCNA in TCGA cohort. (**A**) Clustering dendrograms displayed gene network clustering in high-protein-treated samples, with module diagrams obtained via Dynamic Tree Cut. Merged modules had a heterogeneity coefficient of <0.25; (**B**) analysis of the scale-free fit index (**left**) and average connectivity (**right**) under various soft threshold powers (β = 16); (**C**) eigenvector value correlations were assessed between 25 modules and subtype characteristics; (**D**) the hub gene was calculated from MCC, MNC, and EPC. The darker the color, the more important the gene.

**Figure 4 insects-16-00337-f004:**
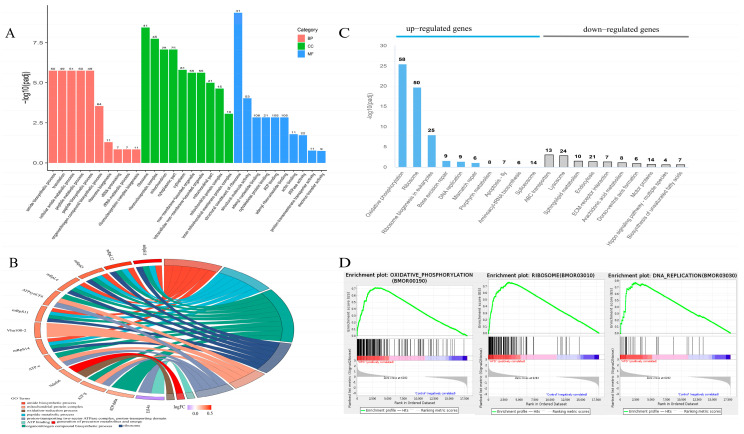
Analysis of GO function and KEGG pathway. (**A**) GO annotation of DEGs. The horizontal coordinate represented the GO term, and the vertical coordinate represented the significance level of GO term enrichment; (**B**) GO chord diagram. On the left was the gene, on the right was the GO term, and the line in the middle indicated the affiliation; (**C**) the KEGG pathway annotation of DEGs. The blue bars represented the up-regulated genes, and the gray bars represented the down-regulated genes; (**D**) gene set enrichment analysis (GSEA); GSEA of several essential pathways in the midgut of silkworms exposed to HPD 6%.

**Figure 5 insects-16-00337-f005:**
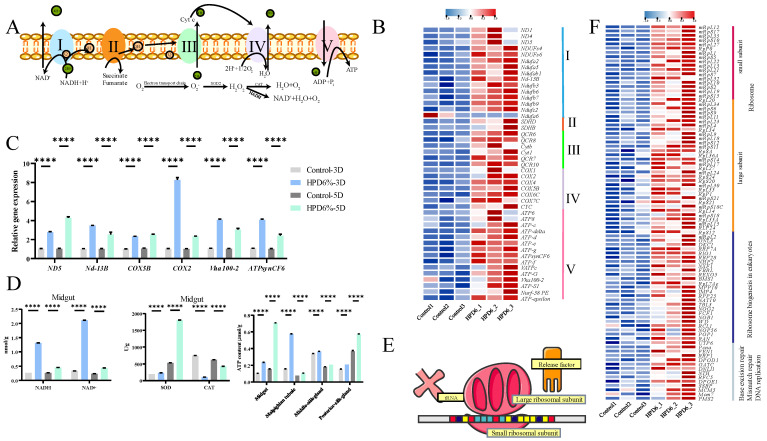
Changes in oxidative phosphorylation and nuclear transcription in the midgut of silkworm. (**A**) Oxidative phosphorylation pathway; (**B**) the expression change of genes in oxidative phosphorylation pathway; (**C**) the expression level of DEGs in oxidative phosphorylation pathway measured by qRT-PCR. “****” indicating an extremely significant difference (*p* < 0.0001); (**D**) the activities of SOD, CAT, NADH, and NAD+ in midgut on I5D3 and I5D5, and the ATP levels in midgut, malpighian tubule, middle silk gland, and posterior silk gland; (**E**) protein synthesis process; (**F**) the expression change of genes in the ribosome, base precision repair, mismatch repair, and DNA replication pathways.

## Data Availability

Data are available in a publicly accessible repository, PRJNA1203685: comparative transcriptome analysis of silkworm midgut on a high-protein diet.

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
