# Peer review of "Comparative Transcriptome Analysis Reveals the Effects of a High-Protein Diet on Silkworm Midgut"

_insects, 2025, doi:10.3390/insects16040337_

Round 1

Reviewer 1 Report

Comments and Suggestions for Authors

Review:

Brief summary:

The manuscript describes research about the effects of “skimmed milk powder” supplemented (3, 6, 12, 18 %) high-protein diet (HPD) on silkworm gene expression. The functions of types/clusters/groups of screened genes were compared/described according to eukaryotes: genes involved in mitochondrial oxidative phosphorylation pathways, ribosome and ribosome biogenesis. The transcriptomic analysis focused on selected gene expression in midgut tissue. The midgut, Malpighian tube, middle- and post- silk glands were tested on NADH and SOD and ATP levels.

1724 differentially expressed genes were identified; the 803 up regulated genes comprised eucaryotic mitochondrial oxidative phosphorylation pathways, ribosome and ribosome biogenesis; the 921 down-regulated genes were annotated as ABC transporters, lysosome, endocytosis and sphingolipid metabolism pathways. The HPD 6% group of silkworms additionally increased DNA repair capacity. ATP levels were increased in midgut, Malpighian tube, middle- and post- silk glands, Activities of NADH and SOD were enhanced in the midgut of HPD 6% group..

The manuscript presents detailed and correct molecular-biology transcriptional analysis. Based on bioinformatical interpretation of results indicating elevated expression of genes involved in metabolism authors claim that HPD 6% promotes silkworm development.

General concept comments

1. Article:

Area of weakness:

The high mortality of control group

Mortality can depend on different factors: strain, season, conditions of rearing (in a laboratory or farm) etc. In the manuscript optimal keeping conditions (i.e., temperature and humidity) are described. Still, for example, for a polyhybrid strain reared in good environmental conditions under springtime temperate conditions, and in the lab, 20% mortality is rather high. Have the larvae shown clinical (subclinical) signs characteristic to certain diseases? Were they tested for dysbiosis or infections?

So, it is unclear why the strain 871 has such high mortality. Why (on basis of which characteristics) have the researchers chosen the 871 silkworm strain for an experiment? Is it a pure, less resistant, high cocoon yield strain or opposite?

It would be nice to know more of such data also in regards of interpretation of (relative?) down-regulated genes for endocytosis, lysosomes, etc.

In reference number (19), describing a silkworm feeding experiment with cow milk enriched high protein diet, the silkworm growth rate (fed with HPD approx. 10-12%) is much higher than in this experiment. How this coulb be explained? In reference (19) scientists additionally describe that there is no difference between mortality of control group and HPD (10 – 12%) "milk" group. In this regards the testability of the hypothesis seems somehow incoherent, including methodology basing on results presented in previous publications.

2. Review:

Based on the comments above, the topics could be additionally supplemented by taking in account that silkworm gut microbiome is strongly (in)directly involved in gut digestion, growth rate, immunological status and consequently gene expression in the gut, etc. Nonetheless, it is hard to combine such huge amount of data in a reasonable way.   

The references from the field of biochemistry, molecular biology and bioinformatics are appropriate. The references from the field of biology (comparison to Drosophila and eucaryotes for example), are somehow driven by force, indicating possible similarities quite generally. There are no references including insect immunology and veterinary medicine. The cited references span comprises last 20 years.

The manuscript is scientifically sound, the experimental design is as much appropriate as general, commercial contemporary approach to similar topics, testing the hypothesis quite generally.

The manuscript is presented in a well-structured manner, considering mostly molecular biology approach and bioinformatical statistical prediction. The weakness of such approach is that the real physiological processes are described very generally.

The manuscript is relevant for the silkworm rearing in quite limited theoretical way, trying to reveal and understand more general patterns of silkworm gene expression (in frames of basic science). HPD based on diluted skimmed milk or similar products are generally not economic for industrial, large-scale silkworm rearing. Additionally, the reeling properties of such cocoons are not known. As the HPD changes the physiological metabolism of silkworms, its use in applied research applications can have unpredictable, adverse effects.

The figures and tables are appropriately supporting the conclusions. The biological interpretation of the statistically supported differences between the analysed parameters of the different silkworm groups in the study should be improved.

Comments on the Quality of English Language

-

Author Response

Comments and Suggestions for Authors:

The manuscript describes research about the effects of “skimmed milk powder” supplemented (3, 6, 12, 18 %) high-protein diet (HPD) on silkworm gene expression. The functions of types/clusters/groups of screened genes were compared/described according to eukaryotes: genes involved in mitochondrial oxidative phosphorylation pathways, ribosome and ribosome biogenesis. The transcriptomic analysis focused on selected gene expression in midgut tissue. The midgut, Malpighian tube, middle- and post- silk glands were tested on NADH and SOD and ATP levels.

1724 differentially expressed genes were identified; the 803 up regulated genes comprised eucaryotic mitochondrial oxidative phosphorylation pathways, ribosome and ribosome biogenesis; the 921 down-regulated genes were annotated as ABC transporters, lysosome, endocytosis and sphingolipid metabolism pathways. The HPD 6% group of silkworms additionally increased DNA repair capacity. ATP levels were increased in midgut, Malpighian tube, middle- and post- silk glands, Activities of NADH and SOD were enhanced in the midgut of HPD 6% group.

The manuscript presents detailed and correct molecular-biology transcriptional analysis. Based on bioinformatical interpretation of results indicating elevated expression of genes involved in metabolism authors claim that HPD 6% promotes silkworm development.

Response: Thanks for the reviewer’s good evaluation and kind suggestion. The advices from you have provided with great helps to us in the current work. Due to your suggestion, we carefully revised our manuscript, If the responses are not accurate and/or in place, please give criticism and more advices. Thank you very much for your careful reviews in our work.

Points 1:Mortality can depend on different factors: strain, season, conditions of rearing (in a laboratory or farm) etc. In the manuscript optimal keeping conditions (i.e., temperature and humidity) are described. Still, for example, for a polyhybrid strain reared in good environmental conditions under springtime temperate conditions, and in the lab, 20% mortality is rather high. Have the larvae shown clinical (subclinical) signs characteristic to certain diseases? Were they tested for dysbiosis or infections? So, it is unclear why the strain 871 has such high mortality. Why (on basis of which characteristics) have the researchers chosen the 871 silkworm strain for an experiment? Is it a pure, less resistant, high cocoon yield strain or opposite? It would be nice to know more of such data also in regards of interpretation of (relative?) down-regulated genes for endocytosis, lysosomes, etc.

Response 1: Thanks for the reviewer’s good evaluation and kind suggestion. We have reared several strains of silkworms under identical environmental conditions. The room is cleaned frequently to maintain a sterile environment, and no strains have shown any obvious signs of disease. Generally, High silk quantity, production and pure breed, the adult rate (observed from 5th-instar-day-1 to the adult period) is 75-80%. The mortality rate of it observed from 5th-instar-day-1 to the adult period is normal in this study. Therefore, we thought that the high mortality rate of 871 due to its genetic traits. The 871 strain is a utilitarian silkworm strain known for its favorable silk production characteristics and is commonly used in hybrid breeding. The mortality rate of it observed from the feeding period to the adult stage, with the control group exhibiting a mortality rate of approximately 20%, which was within the expected range. and this information was supplemented in lines 172-174. Additionally, Endocytosis transforms food into food vacuoles via surface receptors and membrane proteins, and lysosomes fuse with food vacuoles for the digestion of food. Our results indicate that in pathways associated with endocytosis and lysosomes, most down-regulated genes were primarily related to apoptosis and lysosomal acid hydrolases. In contrast, up-regulated genes were enriched in membrane transport proteins and vesicular or vacuolar ATPases. V-ATPase can rapidly modulate the pH of endosomes and/or lysosomes, thereby maintaining the normal functionality of lysosomes. This suggests that increased protein levels may disrupt the lysosome's acidic environment, leading to the regulation of lysosomal acid hydrolases to restore the acidic environment necessary for normal lysosomal function. We added this explanation to the discussion. We have obtained the inspirations and ideas from your strategic and constructive suggestions. Thank you very much.

Points 2:In reference number (19), describing a silkworm feeding experiment with cow milk enriched high protein diet, the silkworm growth rate (fed with HPD approx. 10-12%) is much higher than in this experiment. How this could be explained? In reference (19) scientists additionally describe that there is no difference between mortality of control group and HPD (10–12%) "milk" group. In this regards the testability of the hypothesis seems somehow incoherent, including methodology basing on results presented in previous publications.

Response 2: Thanks for the reviewer’s good evaluation and kind suggestion. It may be caused by the following factors: (1) differences among silkworm varieties, hybrid F1 (BV CSR2 × CSR4) in reference (19), the strain from India, a country located in the tropical region; pure silkworm “871” was used in this study. (2) differences among feeding conditions, “23 ± 1 °C” was used in reference (19); in this study, larvae were reared under standard conditions at 25 -26 °C. (3) differences among treatments, in reference (19), treatments were given on alternate days: “Group 2 was fed with mulberry leaves dipped in milk on alternate days (days 1, 3, and 7) of the fifth instar. On the other days (days 2, 4, and 6), larvae were fed with fresh mulberry leaves”. Our treatment was fed with HPD throughout the fifth instar. (4) The 12% we provided for feeding caused relatively minor damage to the silkworms. However, as the concentration rose, the damage to the silkworms became more severe. In reference 19, feeding is conducted every other day, which is impractical for the application of artificial feed. In the future, our approach might be applicable to the improvement of artificial feed. 

Points 3:Based on the comments above, the topics could be additionally supplemented by taking in account that silkworm gut microbiome is strongly (in)directly involved in gut digestion, growth rate, immunological status and consequently gene expression in the gut, etc. Nonetheless, it is hard to combine such huge amount of data in a reasonable way.

Response 3: We sincerely appreciate the reviewer's insightful comments regarding the significant role of the silkworm gut microbiome in digestion, growth rate, immunological status, and gene expression. These factors are indeed crucial and could provide deeper insights into our research topic. In the current study, our primary focus was on the significant alterations observed in protein metabolism pathways, including oxidative phosphorylation and the ribosomal pathway. However, we acknowledge that the influence of the gut microbiome is an important aspect that warrants further exploration. Given the complexity and volume of data related to the gut microbiome, integrating such information comprehensively is challenging. In this study, we measured the total nitrogen content in silkworm feces to show that protein supplementation is beneficial to the digestive capacity of silkworm. However, the role of microbiota presents a compelling avenue for further investigation, offering valuable insights that could complement our approach. We added this explanation to the discussion. We have obtained the inspirations and ideas from your strategic and constructive suggestions. Thank you very much.

Points4:The references from the field of biochemistry, molecular biology and bioinformatics are appropriate. The references from the field of biology (comparison to Drosophila and eucaryotes for example), are somehow driven by force, indicating possible similarities quite generally. There are no references including insect immunology and veterinary medicine. The cited references span comprises last 20 years.

Response 4: Thanks for the reviewer’s good evaluation and kind suggestion. We sincerely appreciate the valuable comments. We have checked the literature carefully and added more references on 13 and 14 into the INTRODUCTlON part in the revised manuscript. We deleted the original reference 18. This is reflected in lines 61-62. We have obtained the inspirations and ideas from your strategic and constructive suggestions. Thank you very much.

Points 5: The manuscript is scientifically sound, the experimental design is as much appropriate as general, commercial contemporary approach to similar topics, testing the hypothesis quite generally. The manuscript is presented in a well-structured manner, considering mostly molecular biology approach and bioinformatical statistical prediction. The weakness of such approach is that the real physiological processes are described very generally.

Response 5: Thanks for the reviewer’s good evaluation and kind suggestion. Due to your suggestion. We supplemented the supplements the comparison of the size of the silkworm body in line 195-197, and the comparison of the silkworm body size is added in Figure 1. We have obtained the inspirations and ideas from your strategic and constructive suggestions. Thank you very much.

Points 6: The manuscript is relevant for the silkworm rearing in quite limited theoretical way, trying to reveal and understand more general patterns of silkworm gene expression (in frames of basic science). HPD based on diluted skimmed milk or similar products are generally not economic for industrial, large-scale silkworm rearing. Additionally, the reeling properties of such cocoons are not known. As the HPD changes the physiological metabolism of silkworms, its use in applied research applications can have unpredictable, adverse effects.

Response 6: Thank you for recognizing the theoretical value of our paper. The primary objective of this study is to establish a new theoretical framework for sericulture. We appreciate your emphasis on economic value, which we also consider crucial. Currently, this research remains in the theoretical exploration phase. Animal protein sources are abundant and cost-effective, making them a promising nutritional supplement. In cases where mulberry leaf quality is suboptimal, adding an appropriate amount of protein can enhance the conversion efficiency of mulberry leaves into silk. Although our current study does not delve into the economic benefits, these theoretical findings provide a foundation for future improvements in artificial feed formulations. Moving forward, we plan to conduct more applied research to further investigate its economic potential. Thank you again for your valuable feedback.

Points7: The figures and tables are appropriately supporting the conclusions. The biological interpretation of the statistically supported differences between the analysed parameters of the different silkworm groups in the study should be improved.

Response 7: Thanks for the reviewer’s good evaluation and kind suggestion. Due to your suggestion. In line198-201. We added explanations for weight gain of silkworm.

We tried our best to improve the manuscript and made some changes marked in red in revised paper. Thank you very much for your comments and suggestions.

Reviewer 2 Report

Comments and Suggestions for Authors

Chen et al. investigate the phenotypic effects of a high - protein diet (HPD, 6%) on the growth and development of silkworms. Transcriptomic analysis of the silkworm midgut was conducted to identify the mechanisms and metabolic pathways underlying this phenomenon. The oxidative phosphorylation pathway and most ribosomal pathways were analyzed to establish a causal relationship. Moreover, the study provides substantial evidence to support its findings. This includes measurements of SOD and NADH activities in the midgut, as well as analyses of ATP levels in the midgut, Malpighian tubules etc. The study are sound and interesting, as feed addition may be an important direction of sericulture development. However, they are not presented and described well.

Major:

  1. The author fed silkworms with a high - protein diet as an additive and found that both the body weight of the silkworms and the weight of the cocoon shells increased. I think this is reasonable. In the Results and Discussion sections, the author focused on oxidative phosphorylation and nuclear transcription. What I'm more interested in is whether there are also significant changes in nutrition - related signaling pathways, such as the mTOR pathway or the insulin signaling pathway. I think the author should conduct result analysis or discussion in these aspects.
  2. Some abbreviations in the Figure should have their full - length forms provided in the igure legends, such as " 1 2H", "3H" in Fig 1B; In addition, ordinal numbers should be used for expression in Fig 1A, for 1st, 2nd ;

  1. How many silkworms were used for the statistics is not stated in the Materials and Methods section. Additionally, the formula for "cocoon shell rate%" needs to be provided. I think it would be better to include some pictures of silkworms/pupae/cocoons in Fig 1;
  2. The resolution of some figures is insufficient, such as Fig 3 and Fig. 5.

  1. qRT-PCR is mentioned in the Materials and Methods section. Is it also presented in the Results section? Is it shown in sub - figures C and D of Fig. 5 perhaps? It needs to be explained both in the text and in the Figure legend.
  2. The "Conclusions" part doesn't need to be this long and should be made more concise.

The discussion part can increase the comparison with existing studies, especially with the reactions of other insects or model organisms under high protein diet, so as to highlight the innovation and importance of this study.

Check titles of references, and check whether the symbols before quotation in the text are used correctly.

Comments on the Quality of English Language

Minor:

I suggest carefully revising the English. Some sentences are incomprehensible.

Line 55: The paper refers to “The midgut comprises muscle, basilar membrane, epithelium layer, and peritrophic envelope”, where “basilar membrane" should probably be “basement membrane”, and it is recommended to verify the accuracy of the terminology.

Line59-60 “Dietary protein consumption has historically been a limiting element for insect survival, development, and reproductive success", which can be simplified to "Dietary protein is crucial for insect survival, development, and reproduction”. Suggest to simplify the sentence structure to make the expression more clear, some sentences are long and complex, which can easily cause difficulties in understanding. For example,

Line 426-428 “Taken together, our findings provide valuable insights into the wide-ranging effects of treatment in silkworms and expand the current knowledge of the complex biological processes related to molecular mechanisms of nutritional factors in silkworm fifth instar larvae, which can be simplified to“Our findings provide insights into the effects of high-protein diets on silkworms and enhance our understanding of the molecular mechanisms underlying nutritional factors in fifth instar larvae.”

Line 96-98 Improper punctuation of “0(control), 30(HPD3%), 60(HPD6%), 120(HPD12%), 180(HPD18%) and 240(HPD24%) g L-1 skimmed milk powder was used to expose the 5th instar larvae(I5D1).” Statement

 Line 157 The expression of “fluorescence quantitative PCR” is incorrect

Line 159 “BmGAPDH” requires italics

Line 164-165 “Silkworm feces were collected from the HPD and control groups at two and three hours post-feeding on I5D3 and I5D5.” This sentence is not clear

1 Line 213-219 Improper use of italics in Figure 1.

 Line 256 “β= 16” inappropriate use of spaces.

Line 294 The sentence “3.6.hpd6%” needs a space after the punctuation mark

Line 356-358 Syntax problem with “Complex I function as the primary electron acceptor during aerobic respiration chains, facilitating the oxidation of NADH”

Author Response

Response to Reviewer 2 Comments

Comments and Suggestions for Authors:

Chen et al. investigate the phenotypic effects of a high - protein diet (HPD, 6%) on the growth and development of silkworms. Transcriptomic analysis of the silkworm midgut was conducted to identify the mechanisms and metabolic pathways underlying this phenomenon. The oxidative phosphorylation pathway and most ribosomal pathways were analyzed to establish a causal relationship. Moreover, the study provides substantial evidence to support its findings. This includes measurements of SOD and NADH activities in the midgut, as well as analyses of ATP levels in the midgut, Malpighian tubules etc. The study are sound and interesting, as feed addition may be an important direction of sericulture development. However, they are not presented and described well.

Response: Thanks for the reviewer’s good evaluation and kind suggestion. The advices from you have provided with great helps to us in the current work. Due to your suggestion, we carefully revised our manuscript, If the responses are not accurate and/or in place, please give criticism and more advices. Thank you very much for your careful reviews in our work.

Major points

Points 1:The author fed silkworms with a high - protein diet as an additive and found that both the body weight of the silkworms and the weight of the cocoon shells increased. I think this is reasonable. In the Results and Discussion sections, the author focused on oxidative phosphorylation and nuclear transcription. What I'm more interested in is whether there are also significant changes in nutrition - related signaling pathways, such as the mTOR pathway or the insulin signaling pathway. I think the author should conduct result analysis or discussion in these aspects.

Response 1: Response 1: Thanks for the reviewer’s good evaluation and kind suggestion. We found that eIF4E, a crucial protein involved in translation initiation within the mTOR pathway, was upregulated. This suggests that the rate of protein synthesis may have been accelerated due to the increased amino acids derived from milk protein. Therefore, we add a description of the mTOR pathway in line 345-347. However, we did not observe any changes in genes associated with the insulin signaling pathway. We have obtained the inspirations and ideas from your strategic and constructive suggestions. Thank you very much.

Points 2:Some abbreviations in the Figure should have their full - length forms provided in the figure legends, such as " 1 2H", "3H" in Fig 1B; In addition, ordinal numbers should be used for expression in Fig 1A, for 1st, 2nd.

Response 1: Thanks for the reviewer’s good evaluation and kind suggestion. We have changed the full length form of some abbreviations in the figure 1 note, and adopted ordinal representation in Figure 1A. Thank you again for your positive comments and valuable suggestions to improve the quality of our manuscript.

Points 3:How many silkworms were used for the statistics is not stated in the Materials and Methods section. Additionally, the formula for "cocoon shell rate%" needs to be provided. I think it would be better to include some pictures of silkworms/pupae/cocoons in Fig 1;

Response 3: Thanks for the reviewer’s good evaluation and kind suggestion. In Result 3.2, line185 explained the number of silkworms used, and line202-203 provided the formula of cocoon layer rate and the comparison photos of silkworms' body size were also added in Figure 1. Thank you again for your valuable feedback.

Points 4:The resolution of some figures is insufficient, such as Fig 3 and Fig. 5;

Response 4: Thanks for the reviewer’s good evaluation and kind suggestion. We have changed the image to a higher resolution image. Thank you again for your valuable feedback.

Points 5:qRT-PCR is mentioned in the Materials and Methods section. Is it also presented in the Results section? Is it shown in sub - figures C and D of Fig. 5 perhaps? It needs to be explained both in the text and in the Figure legend.

Response 5: Thanks for the reviewer’s good evaluation and kind suggestion. qRT-PCR appears in Figure 5C. We explain this in the diagram notes in Figure 5. Thank you again for your positive comments and valuable suggestions to improve the quality of our manuscript.

Points 6:The "Conclusions" part doesn't need to be this long and should be made more concise.

Response 6: Thanks for the reviewer’s good evaluation and kind suggestion. We have simplified the conclusion. Thank you again for your positive comments and valuable suggestions to improve the quality of our manuscript.

Points 7:The discussion part can increase the comparison with existing studies, especially with the reactions of other insects or model organisms under high protein diet, so as to highlight the innovation and importance of this study.

Response 7: Thanks for the reviewer’s good evaluation and kind suggestion. We have included in the discussion of Line 410-412 a comparison of the responses of other insects or model organisms under a high-protein diet. Thank you again for your positive comments and valuable suggestions to improve the quality of our manuscript.

Points 8:Check titles of references, and check whether the symbols before quotation in the text are used correctly.

Response 8: Thanks for the reviewer’s good evaluation and kind suggestion. We have corrected the formatting errors. Thanks for your careful checks. We are sorry for our carelessness.

Minor points

Points 1:Line 55: The paper refers to “The midgut comprises muscle, basilar membrane, epithelium layer, and peritrophic envelope”, where “basilar membrane" should probably be “basement membrane”, and it is recommended to verify the accuracy of the terminology.

Response 1: Thanks for the reviewer’s good evaluation and kind suggestion. We have revised the technical terms. We sincerely thank the reviewer for careful reading.

Points 2:Line 426-428“Taken together, our findings provide valuable insights into the wide-ranging effects of treatment in silkworms and expand the current knowledge of the complex biological processes related to molecular mechanisms of nutritional factors in silkworm fifth instar larvae, which can be simplified to“Our findings provide insights into the effects of high-protein diets on silkworms and enhance our understanding of the molecular mechanisms underlying nutritional factors in fifth instar larvae.”

Response 2: Thanks for the reviewer’s good evaluation and kind suggestion. We have simplified the summary and modified the statement. Thank you again for your positive comments and valuable suggestions to improve the quality of our manuscript.

Points 3:Line 96-98 Improper punctuation of “0(control), 30(HPD3%), 60(HPD6%), 120(HPD12%), 180(HPD18%) and 240(HPD24%) g L-1 skimmed milk powder was used to expose the 5th instar larvae(I5D1).” Statement.

Response 3: Thanks for the reviewer’s good evaluation and kind suggestion. We have rephrased that sentence. We sincerely thank the reviewer for careful reading.

Points 4:Line 157 The expression of “fluorescence quantitative PCR”is incorrect

Response 4: Thanks for the reviewer’s good evaluation and kind suggestion. We have rephrased that sentence. We sincerely thank the reviewer for careful reading.

Points 5:Line 159“BmGAPDH”requires italics

Response 5: Thanks for the reviewer’s good evaluation and kind suggestion. We've changed the gene name to italics. We sincerely thank the reviewer for careful reading.

Points 6:Line 164-165“Silkworm feces were collected from the HPD and control groups at two and three hours post-feeding on I5D3 and I5D5.” This sentence is not clear

Response 6: Thanks for the reviewer’s good evaluation and kind suggestion. We have rephrased that sentence. And we hope the revised manuscript could be acceptable for you.

Points 7:Line 213-219 Improper use of italics in Figure 1.

Response 7: Thanks for the reviewer’s good evaluation and kind suggestion. We have changed the format. Thanks for your careful checks. We are sorry for our carelessness.

Points 8:Line 256“β= 16”inappropriate use of spaces.

Response 8: Thanks for the reviewer’s good evaluation and kind suggestion. We have changed the format. Thanks for your careful checks. We are sorry for our carelessness.

Points 9:Line 294 The sentence“3.6.hpd6%”needs a space after the punctuation mark.

Response 9: Thanks for the reviewer’s good evaluation and kind suggestion. We have changed the format. Thanks for your careful checks. We are sorry for our carelessness.

Points 10:Line 356-358 Syntax problem with “Complex I function as the primary electron acceptor during aerobic respiration chains, facilitating the oxidation of NADH”.

Response 10: Thanks for the reviewer’s good evaluation and kind suggestion. We have rephrased that sentence. Thank you again for your positive comments and valuable suggestions to improve the quality of our manuscript.

We tried our best to improve the manuscript and made some changes marked in red in revised paper. Thank you very much for your comments and suggestions.

Round 2

Reviewer 2 Report

Comments and Suggestions for Authors

The author has already revised the manuscript according to the suggestions.